# Dynamics of SARS-CoV-2 VOC Neutralization and Novel mAb Reveal Protection against Omicron

**DOI:** 10.3390/v15020530

**Published:** 2023-02-14

**Authors:** Linhui Hao, Tien-Ying Hsiang, Ronit R. Dalmat, Renee Ireton, Jennifer F. Morton, Caleb Stokes, Jason Netland, Malika Hale, Chris Thouvenel, Anna Wald, Nicholas M. Franko, Kristen Huden, Helen Y. Chu, Alex Sigal, Alex L. Greninger, Sasha Tilles, Lynn K. Barrett, Wesley C. Van Voorhis, Jennifer Munt, Trevor Scobey, Ralph S. Baric, David J. Rawlings, Marion Pepper, Paul K. Drain, Michael Gale

**Affiliations:** 1Department of Immunology, Center for Innate Immunity and Immune Disease, University of Washington, Seattle, WA 98109, USA; 2Center for Emerging & Re-Emerging Infectious Diseases, University of Washington, Seattle, WA 98109, USA; 3International Clinical Research Center, Department of Global Health, Schools of Medicine and Public Health, University of Washington, Seattle, WA 98104, USA; 4Department of Epidemiology, School of Public Health, University of Washington, Seattle, WA 98195, USA; 5Department of Pediatrics, School of Medicine, University of Washington, Seattle, WA 98195, USA; 6Division of Allergy and Infectious Diseases, Department of Medicine, School of Medicine, University of Washington, Seattle, WA 98195, USA; 7Allergy and Infectious Diseases Division, Laboratory Medicine & Pathology, & Epidemiology, University of Washington, Seattle, WA 98195, USA; 8Vaccine and Infectious Diseases Division, Fred Hutchinson Cancer Center, Seattle, WA 98109, USA; 9Africa Health Research Institute, Durban 4001, South Africa; 10School of Laboratory Medicine and Medical Sciences, University of KwaZulu-Natal, Mayville 4058, South Africa; 11Centre for the AIDS Program of Research in South Africa, Congella 4013, South Africa; 12Department of Laboratory Medicine and Pathology, University of Washington, Seattle, WA 98195, USA; 13Department of Epidemiology, Gillings School of Global Public Health, University of North Carolina at Chapel Hill, Chapel Hill, NC 27695, USA

**Keywords:** SARS-CoV-2 variant, Omicron, neutralization assay, monoclonal antibody

## Abstract

New variants of SARS-CoV-2 continue to emerge and evade immunity. We isolated SARS-CoV-2 temporally across the pandemic starting with the first emergence of the virus in the western hemisphere and evaluated the immune escape among variants. A clinic-to-lab viral isolation and characterization pipeline was established to rapidly isolate, sequence, and characterize SARS-CoV-2 variants. A virus neutralization assay was applied to quantitate humoral immunity from infection and/or vaccination. A panel of novel monoclonal antibodies was evaluated for antiviral efficacy. We directly compared all variants, showing that convalescence greater than 5 months post-symptom onset from ancestral virus provides little protection against SARS-CoV-2 variants. Vaccination enhances immunity against viral variants, except for Omicron BA.1, while a three-dose vaccine regimen provides over 50-fold enhanced protection against Omicron BA.1 compared to a two-dose. A novel Mab neutralizes Omicron BA.1 and BA.2 variants better than the clinically approved Mabs, although neither can neutralize Omicron BA.4 or BA.5. Thus, the need remains for continued vaccination-booster efforts, with innovation for vaccine and Mab improvement for broadly neutralizing activity. The usefulness of specific Mab applications links with the window of clinical opportunity when a cognate viral variant is present in the infected population.

## 1. Introduction

SARS-CoV-2 was first identified in the western hemisphere near Seattle, WA, USA, in January 2020. Since then, the virus has undergone rapid genetic diversification leading to the progressive outgrowth of new virus strains underlying the pandemic coronavirus infectious disease 2019 (COVID-19) [1]. SARS-CoV-2 has shown remarkable genomic diversification, achieving > 1500 unique Phylogenetic Assignment of Named Global Outbreak Lineages (PANGOLIN) [2]. Some variants have demonstrated increased transmissibility, virulence, and/or immune evasion and are defined as variants of concern (VOC) [3].

The three vaccines approved by the FDA for use in the United States all target the spike protein of the original ancestral SARS-CoV-2 [4]. The neutralizing antibody responses and vaccine effectiveness gradually decline after vaccination and are less effective against emerging VOC [5,6,7,8,9,10]. The mRNA-based vaccine boosters can enhance neutralizing immunity [6,11,12]. Although the viral dynamics of the VOC infections are similar between vaccinated and unvaccinated persons [13], the effectiveness of immunity from previous infection with the non-Omicron virus to prevent reinfection has been substantially lower for the Omicron VOC (56%) compared to the alpha (90%), beta (86%), and delta (92%) variants, indicating differences in immune evasion across these VOC [14]. Consistent with a more extensive immune escape, a study of household SARS-CoV-2 transmission showed higher transmission for Omicron than delta for fully vaccinated adults [15]. These epidemiological results closely match results from immunogenicity studies showing high levels of escape from neutralizing antibody immunity for Omicron subvariants [16,17,18,19,20,21,22,23,24,25,26,27,28,29,30,31].

As humoral immunity is currently the hallmark for protection against SARS-CoV-2 infection, we and others have focused on developing human monoclonal antibodies (Mabs) as leverage in immune therapeutics to treat infection [32,33,34,35]. In particular, therapeutic monoclonal antibodies are useful for treating high-risk and severe COVID-19 patients [36]. We have identified a panel of monoclonal antibodies with neutralizing activity against ancestral SARS-CoV-2 [35]. Here we present a first side-by-side analysis of the extent of humoral immune protection against SARS-CoV-2 variants from convalescent sera following recovery from ancestral virus infection. We also assess novel Mabs for neutralization of emerging SARS-CoV-2 Omicron variants. Our studies reveal a continual need for vaccine applications and the development of Mab therapeutics against contemporary SARS-CoV-2 variants for immune protection against a progressing pandemic.

## 2. Materials and Methods

### 2.1. Isolation of SARS-CoV-2 Variants from Clinical Samples

SARS-CoV-2 specimens were collected by the University of Washington Virology Laboratory. To generate P0 stock, the samples were filtered through Corning Costar Spin-X centrifuge tube filter (CLS8160), and 0.1 mL was used to infect Vero E6 cells ectopically expressing human Ace2 and TMPRSS2 (a gift from Dr. Barney Graham, National Institutes of Health, Bethesda MD) in a 48-well plate. P1 virus stock cultures were produced in Vero E6/TMPRSS2 cells (JCRB1819) and tittered as described [37,38].

### 2.2. Sequencing

An aliquot of P1 stock was subject to RNA extraction (Zymo Research, R1040) and sequencing using the Swift SARS-CoV148 2 SNAP Version 2.0 kit (Swift Biosciences™, Ann Arbor, MI, USA) on Illumina NextSeq 500 (Illumina, San Diego, CA, USA). Consensus genome sequences were generated through a covid-swift-pipeline (https://github.com/greninger-lab/covid_swift_pipeline, accessed on 11 May 2021), and the lineages were assigned based on the Pangolin dynamic lineage nomenclature scheme [39].

#### 2.2.1. Virus Growth Characterization

The growth kinetics of SARS-CoV-2 variants were observed in human lung bronchus epithelial cells, HBEC3-KT (ATCC, CRL-4051), and human lung small airway epithelial cells, HSAEC1-KT (ATCC, CRL-4050), both ectopically expressing human ACE2. 5 × 10^5^ cells were inoculated with designated SARS-CoV-2 virus variants in 250 μL basal medium (PromoCell C-21060, C-21070) at a multiplicity of infection (MOI) of 5 for one-step growth curves, and MOI 0.01 for multistep growth curves. For one-step growth analysis, the virus production was measured at 6, 24, 48, and 72 h post-infection by plaque assay. For multi-step growth, the virus production was measured 1 to 6 days post-infection in quadruplet replicates. Growth curves were generated using GraphPad Prism 9.3.1.

#### 2.2.2. Monoclonal Antibody Expression and Purification

We previously isolated single spike protein receptor binding domain (RBD) specific B cells and sequenced the B cell receptors from three persons. In those studies, we cloned paired heavy and light chain sequences and expressed them as IgG1 monoclonal antibodies, as described [35]. RBD-binding antibodies were further assessed for their capacity to inhibit RBD binding to the ACE2 receptor by surrogate virus neutralization test assay, as described [35,40]. In the present study, we selected a panel of these monoclonal antibodies exhibiting the highest neutralizing activity against ancestral SARS-CoV-2 for analysis of the neutralization of Omicron variants as described below.

#### 2.2.3. Antibody Neutralization Assay

Whole blood samples were collected from participants of multiple study cohorts spanning the duration of the SARS-CoV-2 pandemic from 4/2020–12/2021 (see Tables for details on sera donors). All serum samples were heat inactivated by incubating at 56 °C for 1 h. The protection ability of serum and Mabs against SARS-CoV-2 was assessed by the plaque reduction neutralization test (PRNT) or focus-forming-unit reduction neutralization test (FRNT). Briefly, antibody dilution series were mixed with a virus solution containing 60 pfu for PRNT, or 100 pfu for FRNT. The antibody–virus mixture was incubated at 37 °C for 1 h and then used in a plaque assay with Vero E6/TMPRSS2 cells, as described [37,38], to calculate PRNT50. For FRNT, the antibody–virus mixture was added to 96-well plates pre-seeded with A549-hACE2-TMPRSS2 cells (InvivoGen a549-hace2tpsa, San Diego, CA, USA). At 20 hpi, the cells were fixed and stained using anti-SARS-CoV-2 N (MA5-36086) as the primary antibody, HRP conjugated goat anti-rabbit IgG (ImmunoReagents, GtxRb-003-FHRPX, Raleigh, NC, USA) as the secondary antibody, and TrueBlueTM peroxidase substrate (SeraCare, 5510-0030). The plates were imaged and counted with Cytation 5 (Agilent, Santa Clara, CA, USA). FRNT50 was calculated using GraphPad Prism 9.3.1 by employing non-linear regression analysis. For antibody neutralization of the SARS-CoV-2 encoding a nano-luc reporter gene (SARS-CoV-2-Nluc), antibodies were first diluted serially in a 96-well plate using a Beckman I-5 automation system, then mixed with 800 PFU/mL SARS-CoV-2-Nluc virus per well. The antibody-virus mixture was added to a 96-well plate with each well pre-seeded with Vero cells and incubated for 18 h at 37 °C. Reduction in relative light units (RLU) given by the nanoluciferase reporter was measured using the Promega Nano-Glow Luciferase Assay System, with output captured and recorded on a luminometer. The inhibitory dose (ID_50_) titer for each antibody was determined using a custom Excel macro to process the data set output, and data were graphically processed and visualized with GraphPad Prism, V8, using non-linear regression analysis.

#### 2.2.4. Data Analysis

To compare the antibody titers from cohorts against viral variants, we first evaluated the sample distribution using the Shapiro-Wilk test [41]. The outcomes shown either followed the expected log-normal distribution and were therefore evaluated by paired *t*-test comparing the antibody titers across variants, or were otherwise evaluated by Wilcoxon matched-pairs signed rank test to determine differences in antibody titers between the ancestral virus and Omicron BA.1 variant.

## 3. Results

### 3.1. Isolation of the Variants of Concerns

We established a clinic-to-lab virus isolation pipeline to characterize SARS-CoV-2 variants directly from samples collected at the primary point of care in Seattle, WA, USA. Seattle is geographically placed as where the ancestral SARS-CoV-2 first emerged in the western hemisphere, enabling us to isolate the virus starting from its first emergence through continual variant progression. Figure 1A shows a phylogenetic analysis of a representative isolate of each VOC [39], in relation to SARS-CoV-2/Wuhan-Hu-1 and SARS-CoV-2/WA-1 strains, the letter representing the ancestral North American emergent virus. To evaluate mRNA vaccine-induced humoral immune efficacy across SARS-CoV-2 variants, we first compared the ability of pooled sera from recipients of either mRNA-1273 (n = 8) or BNT162b2 (n = 7) two-dose vaccination regimen, median 8-days prior, against immune sera from convalescent donors having recovered from previous SARS-CoV-2 infection, median 347-days since symptom onset (n = 13) (Table 1), in a PRNT analysis using the live virus of each VOC, prior to the emergence of Omicron BA.1. All VOC tested were effectively neutralized by each vaccination sera pool. Paired *t*-test showed no statistically significant difference between the Moderna and Pfizer mRNA vaccines. In contrast, pooled samples of sera collected from convalescent donors without prior vaccination showed significantly lower neutralization activities compared to either RNA vaccine (*p* < 0.05) across all variants (Figure 1B).

### 3.2. Identification and Characterization of Omicron BA.1

At the beginning of December 2021, we first detected the presence of the Omicron-BA.1 variant in Seattle using the S-gene target failure (SGTF) PCR assay (Figure 2A), following the tail of the delta variant infections. We isolated and sequenced the Omicron-BA.1 variant from these SGTF-identified early cases. By mid-December 2021, the Omicron-BA.1 variant became predominant (Figure 2A, lower). The dominant Seattle Omicron-BA.1 variant is highly similar to the Omicron-BA.1 first reported from South Africa [45], with an additional R346K substitution and four small deletions (Figure 2B). Implementation of 3D modeling of the BA.1 spike protein and spike protein binding to ACE2 placed aa changes in the context of spike protein structure (Figure 2C).

Infection analysis in primary human bronchial epithelial cells (HBEC) and human primary small airway epithelial cells (HSAEC) demonstrated markedly slower replication and significantly lower peak virus production by Omicron-BA.1 (Figure 2D left). Multi-step virus growth analysis revealed slower virus replication and spread by Omicron BA.1 (Figure 2D right), yet exhibited overall better growth in HSAEC compared to HBEC. These results indicate that the Omicron-BA.1 has significantly reduced replication fitness in lung epithelial cells compared to ancestral SARS-CoV-2.

### 3.3. Population Immunity against Omicron BA.1

To assess population immunity, we first evaluated sera from convalescent persons or convalescent persons who received the two-dose vaccine regimen. For this study, convalescence was defined as a median of 157 days after symptom onset for all persons (Table 2). A PRNT50 titer of 10 is the lower-end limit for virus neutralization [48]. We used the neutralization titer geometric mean for each virus to calculate the neutralization differential (ND) value representing the fold increase of neutralization PRNT50 for ancestral virus compared to Omicron-BA.1. Across individual samples, we found that convalescent sera provided low PRNT50 titer (range 1–100) with antibodies that were less effective at neutralizing BA.1 and exhibited the variable ability to neutralize SARS-CoV-2/WA-1 (Figure 3A). However, vaccination with either mRNA-1274, BNT162b2, or J&J-78436735 following convalescence typically provided an increase in PRNT levels that demonstrated enhanced protection against both BA.1 and SARS-CoV-2/WA-1. In non-vaccinated convalescent persons, the mean PRNT50 level was below or at the lower limit for neutralization of either virus, with a nonsignificant trend for enhanced neutralization of ancestral virus compared to Omicron-BA.1. However, in convalescent persons who received any of the three major vaccines, the ND value ranged from 2.8 to 3.5, with significant differences in PRNT50 titer between Omicron-BA.1 and ancestral virus (Figure 3B). As an additional comparator, we included an evaluation of four persons who received a two-dose inactivated virus vaccine regimen of Sinovac [49] or SinoPharm [50] COVID-19 vaccine, with serum collected a median of 165-days post-second vaccine dose (Table 2) [51]. These samples did not show detectable PRNT antibody titer.

### 3.4. Efficacy of Vaccination against Omicron BA.1

We evaluated the efficacy of two- and three-dose vaccination regimens against Omicron-BA.1 in SARS-CoV-2-naïve persons. We conducted paired PRNT50 analyses of sera collected from vaccine recipient cohorts who had the two-dose vaccination, a median of 153 days, or the three-dose vaccination at a median of 32 days after the third vaccine dose (Table 3). This timeline links closely to the emergence of the Omicron-BA.1 variant and the approval for use of the vaccine booster. As shown in Figure 4A, PRNT50 values in persons receiving the two-dose vaccine regimen were consistently lower against Omicron-BA.1 compared to the ancestral virus. In contrast, sera from a three-dose regimen exhibited an increase in PRNT50 against both Omicron-BA.1 and ancestral viruses. Evaluation of mean PRNT50 levels revealed an ND of 20 with the two-dose vaccination. In contrast, vaccination with a three-dose regimen demonstrated enhanced neutralization of BA.1 to nearly completely overlapping levels with ancestral virus neutralization. Remarkably, the ND value of this comparison was reduced to 3.5 compared to ND 20 for a two-dose vaccine regimen (Figure 4B). Comparison of a three-dose to a two-dose vaccine regimen for each virus alone revealed a major enhancement of BA.1 neutralization (ND = 54) over enhancement of ancestral virus neutralization (ND = 9.4) and demonstrated that neutralization of each isolate was significantly improved by the three-dose vaccination.

### 3.5. Efficacy of Monoclonal Antibody against Omicron

We also screened a panel of Mabs for neutralization activity against SARS-CoV-2/WA-1, delta, and BA.1. During the preparation of this manuscript, the Omicron-BA.2 variant emerged in Seattle and was isolated and sequenced. We also obtained Omicron BA.4 and BA.5.2 variants from South Africa. We included each variant in the Mab neutralization assessment. Compared to Omicron-BA.1, the emergent Omicron-BA.2, BA.4, and BA.5.2 variants display additional aa substitutions (shown in Figure 5A). As controls, we compared the Mabs against the clinically used Regeneron REGN10987/10933 two Mab cocktail [36] and a negative control antibody (MaliA01) against a Plasmodium falciparum protein [52]. Within this panel, we identified four Mabs that neutralized the ancestral virus and the delta variant. Remarkably, one of these Mabs (Mab 297) had neutralization activity against both Omicron-BA.1 and Omicron-BA.2 (Figure 5A), although it did not neutralize the BA.4 and BA.5 variants. The Regeneron REGN10987/10933 mAB cocktail had no neutralizing activity against the Omicron variants. We also tested our Mabs against SARS-CoV [53] and bat coronavirus SHC014, with direct comparison to SARS-CoV-2 (Figure 5B), each from infectious clone virus encoding a nano-luciferase reporter [54]. As an additional control, we included the ADG2 Mab, which is known to neutralize all three viruses [55]. Except for the MaliA01 control, all the tested Mabs neutralize the ancestral SARS-CoV-2 but not SARS-CoV. Collectively, these results show that Mab297 can neutralize across the Omicron-BA.1 and Omicron-BA.2, delta, and ancestral viruses to a level comparable to the Regeneron REGN10987/10933 Mab cocktail against the ancestral virus. While REGN10987/10933 does not neutralize the Omicron-BA.1 nor Omicron-BA.2. Mab297 does neutralize these earlier Omicron variants, yet has no neutralization activity against BA.4 or BA.5, which emerged later and more recently in the pandemic.

## 4. Discussion

We isolated every major class of SARS-CoV-2 variant, including all known classes and VOC, except lambda, which has not been known to circulate in the Seattle area. We report the emergence kinetics of the Omicron-BA.1 and its epidemic takeover of SARS-CoV-2 cases from the previously dominating delta variant, followed by the emergence of the Omicron-BA.2 variant. At the time Omicron-BA.1 cases were surging in Seattle, there was about three-quarter of a million cumulative SARS-CoV-2 cases in Washington (WA) state, more than 80% of people in WA have been fully vaccinated, and a certain group of people had the third-dose booster. To assess the population immunity against the emergence of Omicron-BA.1, we compared antibody neutralization of Omicron-BA.1 and SARS-CoV-2/WA-1, showing that sera from convalescent individuals have a reduced ability to neutralize SARS-CoV-2 compared to vaccinated individuals, wherein neutralization activity against Omicron-BA.1 is low or nonexistent. Though our cohort size was small, we show that sera from convalescent individuals who later received Moderna, Pfizer, or J&J vaccination possess variable neutralization activity against Omicron-BA.1 and have an ND of approximately three or greater for neutralization of ancestral virus. Our data clearly show that among SARS-CoV-2 infection-naïve persons who received the two-dose vaccine regimen, the ability to neutralize Omicron-BA.1 was approximately 20-fold lower compared to neutralization activity against the ancestral virus. Importantly, these data show that receiving one vaccine boost clearly closes the gap and reduces the ND between BA.1 and ancestral virus, in addition to improving protective antibody titers against both viruses. In fact, we found that the ND values from the Omicron BA.1 PRNT50 assay were increased by approximately 54-fold when comparing sera from a two-dose versus a three-dose vaccine regimen. These observations along with a related study [56] demonstrate that vaccine boost delivers a significant and functional enhancement in protective antibody levels against both Omicron BA.1 and ancestral virus infection in boosted persons with protective efficacy for 3–4 months [57].

We found that a previously described human Mab, Mab297 [35,58], effectively neutralizes the Omicron-BA.1, Omicron-BA.2, delta variant, and SARS-CoV-2/WA-1. In our assays, the single antibody performed better than the Regeneron REGN10987/10933 mAB cocktail comprising two different Mabs, which do not neutralize these Omicron variants. The ability of Mab297 to neutralize Omicron-BA.1 and Omicron-BA.2 underscores a potential clinical application for this Mab in the treatment of COVID patients. Despite the additional changes in Omicron-BA.1 and Omicron-BA.2, Mab297 possesses neutralizing activity suggesting it binds to epitope(s) conserved across ancestral, delta and Omicron variants. However, newly emerged Omicron variants BA.4 and BA.5 cannot be effectively neutralized by Mab297, demonstrating that these recent variants have evolved to evade immunity developed from ancestral virus infection or vaccination. These observations clearly show that Mab treatment is only effective when the cognate variant is dominant. Strategies to develop pan-variant Mabs [59] should be considered to facilitate Mab-directed therapy for SARS-CoV-2 infection.

Further comparative studies that assessed neutralization of SARS-CoV and SHC014 against bat-derived virus show that Mab297 is not effective against other SARS coronaviruses. In contrast, the positive control of the ADG2 Mab neutralized SARS-CoV-2 ancestral virus and delta variant, SARS-CoV, and SHC014 CoV but not SARS-CoV-2 Omicron variants, indicating that the ADG2 Mab recognizes an evolutionarily conserved epitope on the SARS-CoV-2 RBD overlapping with the hACE2 binding, and suggesting that this epitope is altered in the Omicron variants [60,61].

## 5. Conclusions

Our study provides further evidence affirming that the SARS-CoV-2 Omicron-variants evade neutralizing antibody responses. We show that convalescent sera can neutralize viruses across most major variants of SARS-CoV-2. However, Omicron variants are neither effectively neutralized by convalescent sera nor by standard vaccine regimens. We also present a new monoclonal antibody that has neutralization activity against Omicron strains. Vaccine boosters and/or monoclonal antibodies targeting conserved regions in the viral proteome are warranted for strategies to overcome immunological escape elicited by SARS-CoV-2 variants.

## Figures and Tables

**Figure 1 viruses-15-00530-f001:**
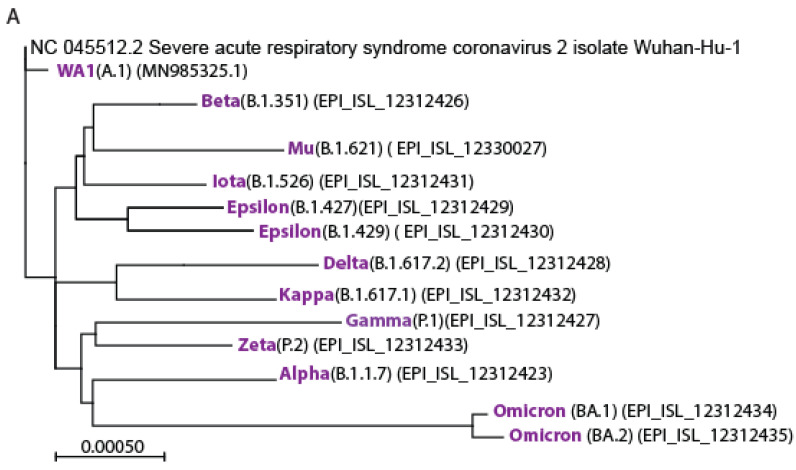
Sequence and neutralization comparison of SARS-CoV-2 variants. (**A**) Phylogenetic tree showing relationships among SARS-CoV-2 variants isolated from patient samples in Seattle, WA, with the SARS-CoV-2/Wuhan-Hu-1 as a reference root. The tree is generated using the online version of MAFFT [42,43] and MEGAX [44]. Tree length is calculated with the maximum parsimony method, the length of the branch is proportional to the number of substitutions. Scale denotes relative genetic distance. (**B**) Geometric mean and standard deviation PRNT50 titer of pooled sera from SARS-CoV-2-naïve individuals vaccinated with Pfizer (BNT162b2) or Moderna (mRNA-12730) vaccine, and patients recovered from prior SARS-CoV-2 infection (convalescent; Table 1). Sera pools were tested multiple times across independent experiments for their neutralization activity against the indicated SARS-CoV-2 variants. The dotted lines show the 10 and 100 PRNT50 reference levels, with PRNT50 of 10 serving as the lower limit cutoff for virus neutralization activity.

**Figure 2 viruses-15-00530-f002:**
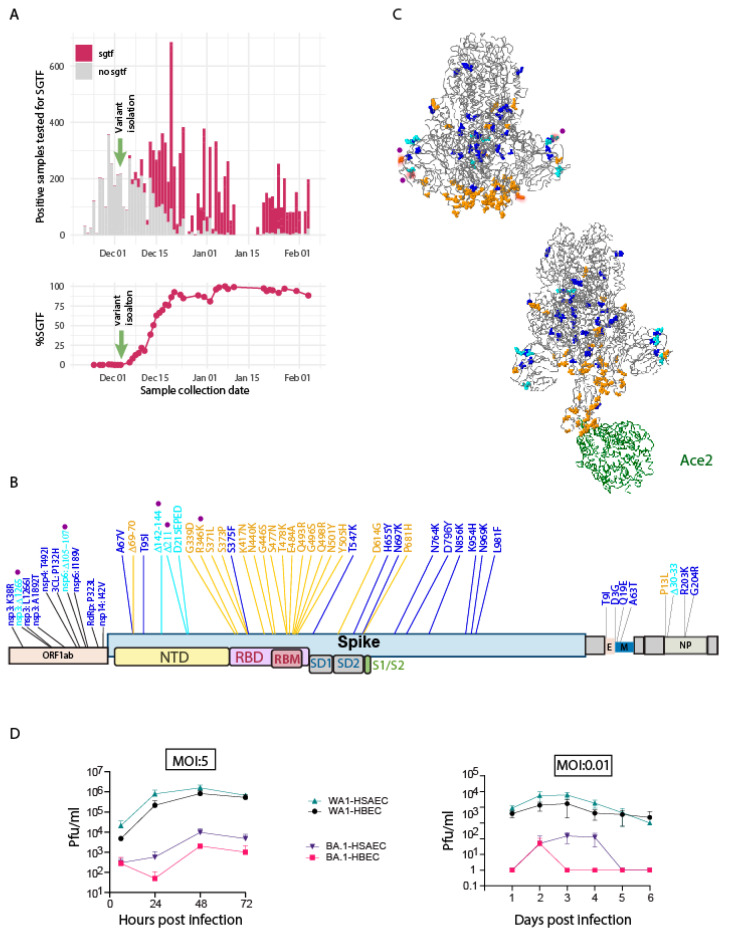
Sequence and characterization of Seattle Omicron strain. (**A**). S gene target failure RT-PCR assay (SGTF; TaqPath assay [46] was applied for new variant surveillance). Upper: Histogram shows the positive cases of Omicron strain emerging in Seattle starting in December 2021 through to February 2022. Lower: Isolation of the Omicron BA.1 variant was isolated and confirmed by viral genome sequencing, replacing other circulating variants for new cases by early January 2022. (**B**). Aa sequence changes in the Seattle Omicron BA.1 variant compared to the ancestral Wuhan-Hu-1 and WA-1 SARS-CoV-2 strains. Purple denotes novel mutations specific to this Seattle Omicron BA.1 variant. Aa changes occurring in more than 100 known Omicron BA.1 sequences appear in blue colored font. If the aa change occurs at a site known to be involved in phenotypic effects, such as altering host-cell receptor binding or antigenicity, it is shown as orange. Aa insertions or deletions are colored in cyan. (**C**). 3D models of Omicron BA.1 spike protein and spike protein binding to ACE2 protein were generated using analysis tools on the GISAID website [47]. Aa color follows the same rule as in B. (**D**). Single-step (left) and multi-step (right) growth and production of infectious virus by SARS-CoV-2/WA-1 and Seattle Omicron BA.1 variant in human lung bronchial epithelial cells (HBEC) and human lung small airway epithelial cells (HSAEC). Plaque titers were measured from culture supernatant over the time course shown. Paired *t*-test was performed, showing that BA.1 produces a significantly lower amount of virus compared to the ancestral strain at 12–48 h post-infection.

**Figure 3 viruses-15-00530-f003:**
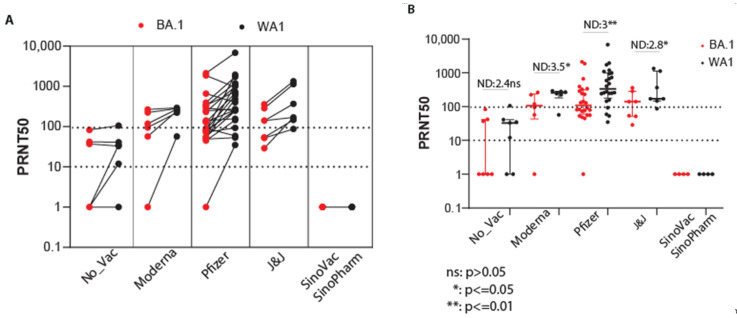
Convalescent sera do not protect against Omicron but vaccination following infection and recovery enhances humoral immunity for virus neutralization. Serum samples were provided from a longitudinal cohort of SARS-CoV-2 infection. Convalescent individuals either received no vaccine or were vaccinated with the indicated vaccines (Table 2). Sera were evaluated for virus neutralization by PRNT50 assay against SARS-CoV-2/WA-1 and the Seattle Omicron BA.1 variant in side-by-side paired testing. (**A**). Geometric mean and standard deviation PRNT50 titers of individual sera in paired testing against each virus. (**B**). PRNT50 of serum analyses showing geometric mean and standard deviation, with all input data points. Neutralization difference (ND) values were calculated by dividing the mean PRNT50 value of SARS-CoV-2/WA-1 neutralization by the PRNT50 mean of Omicron BA.1 neutralization for each sera set. PRNT50 values were statistically evaluated using paired *t*-test. *p*-values are shown for each comparison. The PRNT50 10 lower limit neutralization value and 100 value are marked by the dotted lines.

**Figure 4 viruses-15-00530-f004:**
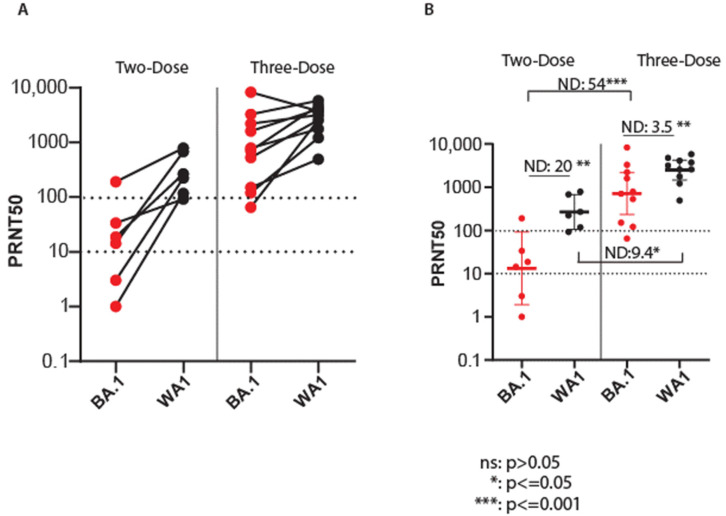
Vaccination/boost regimen enhances neutralization of Omicron. PRNT50 titers were determined from serum samples from SARS-CoV-2 naïve individuals who received the vaccination two-dose regimen (Pfizer vaccine) or three-dose vaccine regimen (Moderna or Pfizer with Moderna boost; Table 3). Neutralization of SARS-CoV-2/WA-1 and the Seattle Omicron BA.1 variant was determined by side paired testing. (**A**). Geometric mean and standard deviation PRNT50 titers of individual sera in paired testing against each virus. (**B**). PRNT50 data of serum analyses showing geometric mean and standard deviation, with all input data points. Neutralization difference (ND) values are shown. PRNT50 values were statistically evaluated using paired *t*-test. Significant differences are indicated with the *p*-value. Ns: nonsignificant. The PRNT50 10 lower limit neutralization value and 100 value are marked by the dotted lines. *: *p* ≤ 0.05, **: *p* ≤ 0.01, ***: *p* ≤ 0.001.

**Figure 5 viruses-15-00530-f005:**
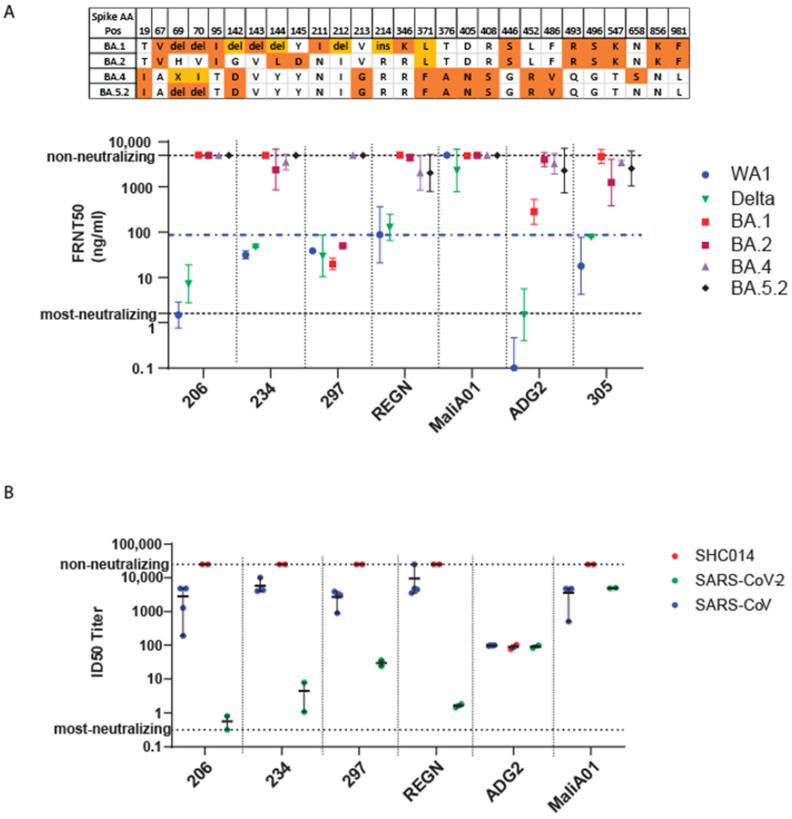
Mab neutralization. (**A**). Upper: Spike protein aa changes comparing Omicron BA.1, BA.2, BA.4, and BA.5 variants. Aa position (AA Pos) in the spike protein is indicated by position number. Del: deletion; ins: insertion. Non-colored (white background) denotes ancestral aa conservation. Colors indicate site variation, with shading marking specific sites of shared changes in multiple variants or unique changes. Lower: Geometric mean and standard deviation FRNT50 in ng/mL Mab concentration shown for the indicated Mabs. 206, 234, 297, and 305 are Mabs that were identified in previous studies for the neutralization of ancestral SARS-CoV-2 [35]. REGN, Regeneron 10987/10933 mAB cocktail. MaliA01 is a negative control antibody against a Plasmodium falciparum protein and is used to establish the non-neutralizing threshold. ADG2 is a positive-control Mab for SARS-CoV. Lower and upper dotted lines mark the FRNT50 levels for most neutralizing and non-neutralizing concentrations, respectively, of the tested Mabs. The middle (blue) dotted line shows the FRNT50 of positive control Mab REGN against WA1. (**B**). ID50 titer analysis of virus neutralization. Mean and range are shown. Lower, middle (green), and upper dotted lines, respectively, mark the ID50 titer level for most neutralizing, ADG2 neutralization of SARS-CoV (positive control), and non-neutralizing thresholds.

**Table 1 viruses-15-00530-t001:** Characteristics of pooled sera sets from vaccinated or convalescent serum.

	Vaccinated (Two-Dose)	Convalescent
**Characteristics**	N = 15	N = 13
Sex		
Male	7 (46.7%)	2 (15.4%)
Female	8 (53.3%)	11 (84.6%)
**Age**		
Min–Max	25–77	22–62
Median (IQR)	35.5 (30–47.8)	45.5 (38–53)
**Vaccine type**		
mRNA-1273	8 (53.3%)	
BNT162b2	7 (46.7%)	
None		13 (100%)
**Time since**	**Recent vaccine dose**	**Symptom onset**
Min-Max	6–20 days	157–366 days
Median (IQR)	8 (8–12.5) days	347 (289.5–360.5) days

**Table 2 viruses-15-00530-t002:** Characteristics of convalescent-vaccination serum.

	Longitudinal Cohort	Inactivated Virus Vaccine Cohort
**Characteristics**	N = 44	N = 4
**Sex**		
Male	25 (56.8%)	3 (75%)
Female	19 (43.2%)	1 (25%)
**Symptomatic**	41 (93.2%)	1 (25%)
**Age**		
Min–Max	19–69	18–21
Median (IQR)	30 (25–46)	19 (18–20.25)
**Pangolin variant lineage**		NA
Alpha (B.1.1.7-like)	10 (22.7%)	
Epsilon (B.1.427/429-like)	1 (2.3%)	
B.1.2	9 (20.5%)	
Gamma (P.1-like)	1 (2.3%)	
Not available	23 (52.3)	
**Vaccine type**		
mRNA-1273	6 (13.6%)	
BNT162b2	24 (54.5%)	
J&J-78436735	7 (15.9%)	
None-convalescent only	7 (15.9%)	
SinoVac		1 (25%)
SinoPharm		3 (75%)
**Time since symptom onset**		NA
Min–Max	177–259	
Median (IQR)	185 (181–190)	
**Time since recent vaccine dose**		
Min–Max	93–177 days	120–188 days
Median (IQR)	130 (112–143) days	165.5 (137.25–188) days

NA: not applicable.

**Table 3 viruses-15-00530-t003:** Characteristics of donors who received a two-dose regimen or three-dose.

	Two-Dose	Three-Dose
**Characteristics**	N = 6	N = 10
**Sex**		
Male	2 (33.3%)	2 (20%)
Female	4 (66.7%)	8 (80%)
**Age**		
Min–Max	31–61	22–63
Median (IQR)	50 (37–56.75)	29 (24–36.75)
**Vaccine type**		
BNT162b2, BNT162b2	6 (100%)	
mRNA-1273, mRNA-1273, mRNA-1273		8 (80%)
BNT162b2, BNT162b2, mRNA-1273		2 (20%)
**Time since recent vaccine dose**		
Min–Max	152–161 days	19–50 days
Median (IQR)	153.5 (153–157)	32 (27–34.5) days

## Data Availability

Primary data from this research is available upon request.

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
