# Peer review of "Dynamics of SARS-CoV-2 VOC Neutralization and Novel mAb Reveal Protection against Omicron"

_viruses, 2023, doi:10.3390/v15020530_

Round 1
Reviewer 1 Report
Linhui Hao et al. present an analysis of the extent of humoral immune protection against SARS-CoV-2 variants. They isolated major SARS-CoV-2 variants that used to circulate in the Western hemisphere. Variants were rapidly isolated, characterized and used to evaluate humoral immunity from convalescent sera or vaccination. In addition, they assess the effectiveness of several mAbs that they developed previously against VOC including Omicron sub-variants. However, too much similar work has been published because the immune escape features of novel variants are a major global concern and super hot topic. Despite the effort, there is little new discovery in this work.
1. Lines 98-103 It seems that the mAbs were isolated in previous studies. This section should be “Monoclonal antibody expression and purification”;
2. Lines 203-204 Figure 2C. Since several cryo-EM structures of Omicron BA.1 spike protein have been published, the 3D model could be updated to a real structure.
Author Response
- Lines 98-103 It seems that the mAbs were isolated in previous studies. This section should be “Monoclonal antibody expression and purification”; Response: We have changed the title of this section accordingly.
- Lines 203-204 Figure 2C. Since several cryo-EM structures of Omicron BA.1 spike protein have been published, the 3D model could be updated to a real structure. Response: We appreciate this comment, but have not change this figure, as we are currently working on the Cryo-EM structure for the BA.1 variant that we isolated. The 3D model here is a computer simulation based on the sequencing information and prior structures, and we feel that it sufficiently shows the information that we are trying to convey.
Reviewer 2 Report
Comments to the Author
In this study, a clinic to lab viral isolation and characterization pipeline was established to rapidly isolate, sequence, and characterize SARS-CoV-2 variants. This paper introduce the virus neutralization assay between SARS-CoV-2 variants and antibody from infection or vaccination. The result showed that convalescence greater than 5-months post-symptom onset from ancestral virus provides little protection against SARS-CoV-2 variants. A novel Mab297 neutralizes Omicron BA.1 and BA.2 variants better than clinically approved Mabs but neither can neutralize Omicron BA.4 or BA.5.
This study is valuable because it proved SARS-CoV-2 Omicron-variants evade neutralizing antibody responses. And, it provides direction for vaccine research. However, there are some problems and deficiencies that need to be solved in the article. These are my concerns on specific pages and parts:
1. In Figure 3, the drawing format of Figure 3A should be consistent with other figures in the article.
2. In Table 2, there is a large difference in the number of longitudinal cohort and inactived virus vaccine cohort. The author should augment the number of inactived virus vaccine cohort.
3. In Figure 2, the 3-D modeling of the BA.1 spike protein and spike protein binding to ACE2 (Figure 2C, lower) is not clearly explained.
4. I think the conclusion of this article does not show the whole work of the study well, the conclusion is not detailed, thus, it can be introduced in more detail.
5. The author does not explain how to screen the monoclonal antibodies, and the author needs to describe the screening method in detail.

Author Response
- In Figure 3, the drawing format of Figure 3A should be consistent with other figures in the article. Response: We have modified the drawing format of Figure 3A to match the style with other figures.
- In Table 2, there is a large difference in the number of longitudinal cohort and inactivated virus vaccine cohort. The author should augment the number of inactivated virus vaccine cohort. Response: We appreciate this comment. We now have stated these differences as a possible limitation to our study.
- In Figure 2, the 3-D modeling of the BA.1 spike protein and spike protein binding to ACE2 (Figure 2C, lower) is not clearly explained. Response: 3D models of Omicron BA.1 spike protein and spike protein binding to ACE2 were generated using analysis tool on the GISAIS website [61]. This information is now included in the legend of Figure 2.
- I think the conclusion of this article does not show the whole work of the study well, the conclusion is not detailed, thus, it can be introduced in more detail. Response: We agree, and we now present more details in our conclusions.
- The author does not explain how to screen the monoclonal antibodies, and the author needs to describe the screening method in detail. Response: We now add additional text in the Methods section to better explain the antibody screening process and our workflow in this study.
Reviewer 3 Report
In this manuscript, Hao et al., compared the neutralizing Abs titers elicited by vaccines against SARS-CoV-2 variants. In addition, they also used mAb which they previously developed. They showed that booster shot increased its neutralizing activity against current Omicron variants. They also found that one of the their mAb termed 297 efficiently neutralized BA.1 and BA.2. Overall, the experiments and analysis are well performed. I have a few comments for this manuscript.
1. Figure 4 title: The authors described “Vaccination/boost regimen enhances protection against Omicron”. However, Figure 4 just showed vaccination/boost enhanced neutralizing titers, not protection. The authors should change the title.
2. Figure 5: there are no explanation about Mab206, 234 and 305. Please add the explanation about these Abs.
3. Figure 5A lower: what the blue dot line means? Please add the description in the Figure legend.
4. The authors should expand the introduction to explain about the neutralizing activity of therapeutic mAbs and their Abs (Mab297, 206, 234 and 305) against parental and variants.
5. The sample size is small to conclude. Please add the limitation of the study section in the “Material and Methods”.
Author Response
- Figure 4 title: The authors described “Vaccination/boost regimen enhances protection against Omicron”. However, Figure 4 just showed vaccination/boost enhanced neutralizing titers, not protection. The authors should change the title. Response: We have changed the Figure 4 title accordingly.
- Figure 5: there are no explanation about Mab206, 234 and 305. Please add the explanation about these Abs. Response: We have now added these details to the legend of Figure 5.
- Figure 5A lower: what the blue dot line means? Please add the description in the Figure legend. Response: The blue dotted line shows the FRNT50 level of neutralizing concentration for positive control Mab REGN against WA1. This is now clearly stated in the legend.
- The authors should expand the introduction to explain about the neutralizing activity of therapeutic mAbs and their Abs (Mab297, 206, 234 and 305) against parental and variants. Response: We have added additional text in the introduction describing the monoclonal antibodies in that they were leveraged in the present study for testing against Omicron variants.
- The sample size is small to conclude. Please add the limitation of the study section in the “Material and Methods”. Response: We have added the following sentence to the Institutional Review Board Statement paragraph: “In addition, we acknowledge that the smaller size of our cohorts could be a limitation to this study.”
Round 2
Reviewer 2 Report
The author has solved the problems that appear in the article, the manuscript has been sufficiently improved to warrant publication in Viruses.